# The Metagenomic and Metabolomic Profile of the Infantile Gut: Can They Be “Predicted” by the Feed Type?

**DOI:** 10.3390/children9020154

**Published:** 2022-01-25

**Authors:** Eftychia Ioanna Dimitrakopoulou, Abraham Pouliakis, Vasiliki Falaina, Theodoros Xanthos, Panagiotis Zoumpoulakis, Thalia Tsiaka, Rozeta Sokou, Zoi Iliodromiti, Theodora Boutsikou, Nicoletta Iacovidou

**Affiliations:** 1Department of Neonatology, National and Kapodistrian University of Athens, 157 72 Athens, Greece; sokourozeta@yahoo.gr (R.S.); ziliodromiti@yahoo.gr (Z.I.); theobtsk@gmail.com (T.B.); niciac58@gmail.com (N.I.); 22nd Department of Pathology, National and Kapodistrian University of Athens, 124 62 Athens, Greece; apou1967@gmail.com; 3Neonatal Intensive Care Unit, General Hospital of Nikaia, Piraeus “Agios Panteleimon”, 184 54 Piraeus, Greece; vickyfalaina@gmail.com; 4Department of Medicine, European University Cyprus, Nicosia 2404, Cyprus; theodorosxanthos@yahoo.com; 5Institute of Chemical Biology, National Hellenic Research Foundation, 116 35 Athens, Greece; pzoump@eie.gr (P.Z.); tsiakath@uniwa.gr (T.T.); 6Department of Food Science and Technology, University of West Attica, 122 43 Egaleo, Greece

**Keywords:** infant, feed type, metagenomics, metabolomics, fecal microbiota, gut chemistry, prebiotics, supplemented formula

## Abstract

Purpose: The composition and the metabolic activity of the gut microbiota of breastfed and formula-fed infants has been the focus of several studies over the last two decades. Gene sequencing techniques and metabolomics in biological samples have led to expansion of our knowledge in this field. A more thorough comprehension of the metabolic role of the intestinal microbiota could assist and expedite the development of optimal feeding strategies. The aim of this systematic review is to present available data regarding the effect of the feed type on the infantile intestinal microbiota (microbial composition and metabolites) by DNA-sequencing and metabolome analysis of neonatal stool. Methods: A systematic search of the literature in PubMed was attempted to establish relevant studies. Randomized controlled trials studying the diversity and composition of gut microbiota and metabolites of infants that received different types of feed were included. The study subjects were infants/neonates born at term or preterm receiving either breast, donor, or formula milk. Formula could be either classic or fortified with probiotics, prebiotics, or both. The included trials compared the differences on metagenomics and metabolomics of infantile stool, aiming at investigating the beneficial effects of fortification of formula with synbiotics. Results: Out of 1452 papers identified by the initial search, seven were selected for inclusion, following screening for eligibility. Eligibility was determined by closer examination for relevance of the title, abstract, and subsequent full text. The results of these studies mostly support that the feed type modulates the microbiome composition. In terms of the alpha-diversity, no significant difference exists between the feeding groups, whereas significant differences were noted with regards to beta-diversity in breastfed and formula-fed infants. As for the microbial composition, the studies revealed different populations in the formula-fed group compared to the breastfed group at the phylum and genus level. Bifidobacteria supplementation of infant formula did not seem to change the proportions of Bifidobacterial sequences during the first year of life. Another finding according to the studies is that the pH of fecal samples in breastfed as well as prebiotic-supplemented formula-fed infants. was significantly lower than that of formula-fed infants. Infant milk formula with a mixture of prebiotics (GOS/FOS oligosaccharides) was shown to be capable of selectively stimulating the growth of Bifidobacteria with analogous changes in fecal pH and short-chain fatty acid content in fully formula-fed infants. Conclusions: Overall, there is evidence to support that feed type modulates the infants’ microbiome constitution. The impact of feeding on term and preterm microbiota could have potential benefits on intestinal functionality, immune system, and metabolism, and probably pursuing the host for life.

## 1. Introduction

Breast milk is the most appropriate source of nutrition for infants. In addition to normal nutrients, such as fats, proteins, carbohydrates, vitamins, and minerals, breast milk carries many more biologically active molecules, including growth factors, cytokines, antimicrobial substances, and immune-enhancing components. The dietary advantages of breastfeeding and its defense against infection have been well demonstrated [1,2].

When breast milk is not available, formula milk will substitute. Many efforts have attempted to create a formula as close as possible to breast milk’s composition, aiming at optimizing the infant’s growth and development [3].

Over the last few years, it has become clear that crucial imprinting events are modulated in early-life by nutrition, which contributes to potentially long-lasting effects for the infant [4]. These events might be mediated directly or through changes on the infant microbiome. The power of the microbiome to control and alter host responses in infancy depends on individual bacterial species, which modulate metabolic responses, immune regulation, adipogenesis, and possibly even brain maturation and cognitive function [2,5,6]. Abnormal early-life colonization of the intestinal tract has a detrimental life-long impact on immune regulation and metabolic homoeostasis [2].

The composition and the metabolic activity of the gut microbiota of breastfed and formula-fed infants has been the focus of several studies over the last two decades.

Gene sequencing techniques, and metabolomics in biological samples, such as breast milk, urine, or stool, have led to expansion of our knowledge in this field. The capacity to bring to light the identities, activities, and functionalities of the gut microbiota is an instrument of utmost importance.

Several authors reported that fecal samples of breastfed infants consist of higher amounts of Bifidobacteria and Lactobacilli, while they contain lower levels of potential pathogens compared to that of formula-fed infants. Furthermore, breastfed newborns carry a more stable and invariable microbial population when compared to the formula-fed ones [7], which acquire an intestinal microbiota with Staphylococci, Bacteroides, Clostridia, Enterococci, Enterobacteria being the mainly abundant members [8]. As a result of these differences, the levels of specific metabolite classes, such as lipids, hormones, and amino acids, are also divergent in the stools of breastfed versus formula-fed infants, with the presence of propionate and butyrate being higher in the latter group [8,9]. In addition, it seems that formula-fed infants attain an early microbial diversity toward an adult-like gut colonization [10].

A more thorough comprehension of the metabolic role of the intestinal microbiota, and of the profile and dominant participants in the gut microbial population hold considerable promise to assist and expedite the development of optimal feeding strategies [11]. Formulas with the addition of probiotics, prebiotics or both were produced to enforce the growth of Bifidobacteria. This formula enrichment was also expected to promote and improve the metabolic activity of the intestinal flora in total. Evidence in the literature suggests that the consumption of infant formula fortified with prebiotics and/or probiotics leads to the development of a neonatal gut microbiota that resembles that of breastfed infants [7,12].

The aim of this systematic review is to present available data with an objective, reproducible method of investigation of the effect of the feed type on the infantile intestinal microbiota (microbial composition and metabolites) by DNA-sequencing and metabolome analysis of neonatal stool. This review also aims to evaluate whether clear differences, depending on feed type, exist, and how robust these findings are.

## 2. Materials and Methods

In order to find existing research outcomes that analyze and prove how feed type modulates the infants’ microbiome and metabolome constitution, available studies related to this specific issue were reviewed, and subsequently, the reported results were combined and compared.

A systematic literature search via PubMed was attempted and relevant studies were identified. Publications over the last 20 years, were retrieved with the help of combined keywords and was identified original research investigating the differences on the metagenomic and metabolomic profile of feces of infants fed exclusively with breast milk compared to that of infants fed with formula/fortified with synbiotics.

No exclusion criteria were applied regarding gestational age or postnatal period of observation of the study populations. Studies and general reviews involving the analysis of biological specimen other than stool were excluded from the assessment. Studies not including an analysis of the metabolites and the composition the intestinal microbiota depending on the feed type were also excluded in the process of the initial screening of the abstracts and papers.

The present study was performed in strict accordance with guidelines for meta-analysis. Eligible studies published on PubMed, up to 15 September 2020 (the study data collection time instance) were selected and were considered eligible for a potential inclusion in his systematic review.

We selected trials studying the diversity and composition of gut microbiota and metabolites of infants that received different types of feed. The study subjects were infants/neonates born at term or preterm receiving either breast, donor, or formula milk. Formula could be either classic or fortified with probiotics, prebiotics, or both. The time of postnatal observation differed in each study. Infant stool was the biological specimen investigated. Trials compared the differences on metagenomics and metabolomics of infantile stool, aiming at investigating the beneficial effects of fortification of formula with synbiotics. Primary outcome measures were the identification of microbial composition and metabolites in the gut and the recognition of any differences on the population and metabolic activity depending on the feed type. Secondary outcome measures were the fecal pH, SCFA (short-chain fatty acids), lactate, and occasionally growth-anthropometrics and stool characteristics.

The search query was formulated according to the PICO framework [13,14]. PICO is a framework used in evidence-based medicine to create, in a systematic manner, clinical or health related questions, and moreover, to create search strategies for literature research. PICO stands for: P: patient, population, or problem; I: intervention; C: comparison or control; and O: outcomes. According to this strategy, the formation of individual query parts and the final query issued in PubMed is depicted in Table 1. Mesh terms were extensively used to be compliant as much as possible to the standard practice; however, the keywords were allowed to be in the abstract/title of the publications.

The query was issued to the PubMed database using the advanced search builder, which allows the issuing of handwritten questions. The individual parts of the query were used for specific searches: the combination with the AND operator of the above components resulted in the final query (Table 1), which returned 1452 potentially eligible research papers.

The procedure that was followed for selecting the relevant literature is summarized in Figure 1. Each abstract and subsequent individual papers were screened independently by two individuals for eligibility. In the event of conflicts of opinion, a discussion was performed in order to achieve a consensus. The PRISMA [15] approach was applied for this review.

In the course of the systematic assessment, study data were reviewed, whenever possible, in relation to gestational age and number of groups of the participants, feed type, techniques used for analyzing the specimens, and differences reported in the composition and metabolic consequences between the infants.

A table of data extraction was developed to ensure that the same information and variables were primarily collected from the included studies, including the type of intervention, study subjects, follow-up duration, applied methods, and the reported outcomes of the studies. The collected information was subsequently checked by the same two authors that performed the review of the papers.

With regards to the risk of bias in the individual studies and the overall quality of the body of evidence, we followed the GRADE approach [16] (more details are presented in Appendix A).

## 3. Results

A total of seven studies, including five randomized controlled trials; one prospective, observational cohort study; and one secondary analysis of data from a prospective exploratory study, were distinguished for inclusion in the review. The studies that were excluded were those not relevant mainly because they used different biological specimens or the study population other than neonates. A search of PubMed databases was conducted that provided a total of 1452 outcomes at first. Of these, 1366 studies were excluded because after reviewing the titles, these papers appeared not to be as relevant. Sixty-six additional studies were also excluded because after reviewing the abstracts, it appeared that these papers evidently did not meet the criteria. The full text of the remaining 20 citations was studied in more detail. It turned out that 13 studies failed to meet the inclusion criteria, as described, because they examined the metabolome in urine, or milk samples, or the studies were not conducted on humans. Seven studies met the inclusion criteria in terms of participants, type of intervention, and aim of the study, thus were included in the systematic review.

The duration of the intervention was variable from during the first month of life to 24 months of age. Although one study lasted for 24 months, ultimately the observation was only close during the first 12 months, with only some of the participants attending the 2-y follow-up. The included studies involved variable sizes of populations. For the study subjects, the main inclusion criteria entailed term and preterm neonates and infants, fed with either breast milk or formula milk (standard or supplemented); the intervention always occurred after obtaining parental consent before study entry. Three of the trials were multicentric. In all studies, the primary outcome was gut microbial patterns associated with feed type and in one study, the primary outcome was SIgA. All studies evaluated confounding factors, including the effect of gestational age, the mode of delivery, use of antibiotics, pathology of the mother, and the presence of co-morbidities. Secondary and additional outcomes were fecal pH, SCFA (short-chain fatty acids), lactate, and also growth-anthropometrics and stool characteristics in some articles. The timing of outcome measures was variable and could include monthly evaluations and sampling for a year, or evaluations every two weeks for one month.

Exclusion criteria included congenital abnormalities, surgery, maternal chronic illness, and antibiotics received either by the infants or the mothers.

Breastfed infants comprised the control group. The specimen used in all studies was frozen stool samples; storage conditions in relation to storage temperature and storage time length before analysis of fecal samples varied. Fluorescent in situ hybridization (FISH) was used for bacterial detection in two studies, 16S rRNA sequencing was used in another two studies, and three studies used 16S rRNA gene amplified by PCR in analyzing the samples.

Different reference databases were used for the assignment of taxonomy, such as QIIME (Quantitative Insights Into Microbial Ecology), UPARSE, Greengenes 16S rRNA gene database, as well as alpha diversity indices (Chao1 and Shannon), beta diversity (UNIFRAC and Bray–Curtis), and statistical methods to analyze possible discrepancies between study population or controls concerning their gut microbial community properties, with non-parametric multivariate statistical tests, such as PERMANOVA (permutational multivariate analysis of variance).

More specifically, in one study, two groups of infants fed infant formula with GOS/FOS and a control group fed with standard formula were placed in a randomized, double blind, placebo controlled intervention study. Simultaneously, a breastfed group was evaluated. At study onset and after 4 and 6 weeks, fecal samples were collected and studied for the amount of Bifidobacteria, as well as biochemical factors, such as pH, short chain fatty acids, and lactate.

The second study included preterm newborns that were enlisted at birth and followed up for 30 days. Infants were classified into six groups (maternal milk, donor milk, formula, and then three more groups that were mixed-fed either with maternal and donor milk or with formula and one of the two). Stool samples were collected every day during the first month of life.

The next study included infants that were observed from birth to 12 months postnatally and they were separated into three groups: breastfed, standard formula-fed, and fortified-formula-fed (probiotics).

In the next study that was included, the impact of an infant fermented formula also containing a mixture of prebiotics was studied in healthy term infants. Three experimental formulas either containing a combination of bioactive compounds (FERM) and prebiotic oligosaccharides (FERM/scGOS/lcFOS), prebiotic oligosaccharides (scGOS/lcFOS), or the bioactive compounds (FERM) were compared to a standard control formula. As a reference, healthy term infants who received breast milk exclusively during the first six months of life were surveyed.

In the other study, the purpose was to explore the differences on the initial composition and gene expression of gut bacteria depending on the feed type in moderate late-preterm infants. The two groups included babies feeding with maternal milk and those feeding with formula.

The next study evaluated the impact of infant feed type from birth until 12 months of life depending on the duration of breastfeeding and the timing of formula introduction. Infants were classified into three groups: the exclusively breastfed group, infants who had been only fed with formula and infants who were on mixed feeding (both with breast milk and formula) prior to their stool sample.

The last study had the objective to assess the impact of donor human milk upon preterm intestinal microbiota. The population was neonates with gestational age < 32 weeks and with a birth weight ≤ 1500 g. Neonates were classified in three groups according to feeding practices consisting of their maternal milk, donor human milk, or formula.

All the studies included analyzed gut microbiota composition and metabolites of neonatal stool samples and compared the differences between different groups depending on the feed type.

Although the studies were similar regarding the data and the outcomes presented, a meta-analysis was not performed as evaluation showed it would increase the risk of bias. The studies that are presented show discrepancies in the size of the population or the structure of the methods and results The use of different analysis, design, and time-points of assessment causes difficulties combining the data using a statistical process and the individual research outcomes cannot be combined quantitatively. The analysis was limited to a systematic qualitative review; the outcomes and characteristics of the separate studies are reported in a descriptive form (Table 2).

The results of these studies are mostly in line with the main findings that the feed type modulates the microbiome composition.

### 3.1. Alpha— and Beta—Diversity

In terms of the intrasample bacterial richness and diversity (alpha-diversity), little or no significant difference exists between the feeding groups, whereas with regards to the intersample, microbial variations (beta-diversity) in breastfed and formula-fed infants were significantly different. In three studies, the bacterial abundance and diversity proved lower in breastfed infants vs. formula-fed infants [18,21,22]. Only in one study, alpha-diversity of the gut microbial population was higher in the maternal-milk group, and that referred to a specific population of preterm neonates [17]. It is also notable that the included intervention study [18] indicated that formula enriched with *Bifidobacteria* does not affect intestinal a- and b-diversity to a significant extent.

### 3.2. Microbial Composition

As for the dominant phyla that comprise the gut microbiome, the studies revealed that *Firmicutes* was the predominant bacterial phylum in the majority of infants of both breastfed and formula-fed groups, although in one study on preterm infants significantly higher relative abundance of *Firmicutes* was noticed in formula-fed group compared to breastfed group.

In terms of genus, the *Escherichia* and *Clostridium* and the family of *Enterobacteriaceae* seem to be more present in the formula-fed groups. Among the breastfed infants higher abundance of *Bifidobacteriaceae* was observed, mostly in term, and lower abundance of *Clostridiaceae* [12,18,19,20,22]. In one study with late-preterm infants [21], no difference was found in the relative abundance of *Bifidobacterium* and *Lactobacillus* (both considered to be advantageous and mainly colonizing the gastrointestinal tract of full term healthy infants) between breastfed and formula-fed groups. The main health-relevant genera in all groups are *Bifidobacterium* and *Lactobacillus*.

Li et al. [22] showed that the percent composition of *Bifidobacterium* and *Lactobacillus* species relative to other species was higher in the breastfed group in comparison to other groups. *Propionibacterium* on the other hand, as reported by Wang et al. [21], had remarkably greater relative abundance in the breastfed group than in the formula group, and specifically was found to be the most informative in effectively predicting the separation of the two groups. Formula-fed groups in different studies were associated with a significantly higher relative abundance of *Bacteroides* and one *Blautia* species [12,18,22]. Bazanella et al. [18] reported that supplementing infant formula with *Bifidobacteria* significantly weakened the relative abundance of *Bacteroidaceae* at 1 month of age, but these differences largely disappeared at 3 months, supporting the overall finding of the study that bifidobacterial intervention does not show any long-term aftermath on fecal microbiome.

*Bifidobacteria* supplementation of infant formula also did not seem to change the proportions of bifidobacterial sequences during the first year of life. Such an intervention is therefore unlikely to compensate for differences in microbiota composition observed between breast- and formula feeding as the supplementation with *Bifidobacteria* alone was inadequate in altering the microbial populations and the stool metabolite profile over a longer period of time in this study [18].

Knol et al. [12] reported that, although the baseline values for the percentage of *Bifidobacteria* in the prebiotic-supplemented (GOS/FOS oligosaccharides) formula group and the standard-formula group were similar, after 6 weeks the percentage of *Bifidobacteria* increased in the prebiotic-supplemented formula group compared to the standard formula group.

### 3.3. Gut Chemistry and Metabolites

Another factor that appeared to play a role in affecting neonatal gut microbiome is the acidity of the gut, resulting from nutritional sources. According to the studies, the pH of fecal samples in breastfed, as well as prebiotic-supplemented formula-fed infants was significantly lower than that of formula-fed infants [12,20,21].

Knol et al. [12] demonstrated as well, that an infant milk formula with a mixture of prebiotics (GOS/FOS oligosaccharides) is capable of selectively stimulating the growth of *Bifidobacteria* with analogous changes in fecal pH and short-chain fatty acid content in fully formula-fed infants. This study demonstrates the enhanced growth of *Bifidobacteria* and a related impact on metabolic activity in infants with an already formed gut microflora associated with conventional milk formula. Microbial enzymatic decomposition of non-digestible carbohydrates and proteins in the colon causes a decreased luminar pH mainly attributed to the generation of short-chain fatty acids and lactate.

The short-chain fatty acid (SCFA) pattern of infants in the prebiotic-supplemented formula-fed infants, approximates the pattern of the short-chain fatty acids of breastfed infants. High levels of acetate, and low levels of propionate and butyrate appeared to arise following feeding a standard formula or breast milk. The distinct SCFA profile that appeared between the groups of infants fed the prebiotic mixture and those fed the standard formula, is an indication that the composition of the microflora differs between the two groups. This was consistent with the finding of a higher percentage of *Bifidobacteria* in the prebiotic-supplemented formula-fed group compared to the standard group. The production of acetic acid and lactic acid has been attributed to *Bifidobacteria* and lactic acid bacteria in the colon of breastfed infants [12].

The findings of Li et al. [22] supported that *Bifidobacterium* was remarkably, positively related to specific metabolites, such as L-proline, D(–)-arginine, DL-threonine, while there was a negative correlation to creatine, capric acid, and taurine. An increase in proline seemed to result from a relatively large community of *Bifidobacterium*. Specific metabolites were also associated with *Lactobacillus*, *Bacteroides*, or *Klebsiella*. The amino acid metabolic pathway was more related to breastfed infants supporting that the amino acid synthesis pathways are increased in the microbiota of these infants. Compared with the breastfed group, the mixed- and formula-fed groups were more related to fatty acid biosynthesis and biosynthesis of unsaturated fatty acid [22]. On the other hand, the study of Parra-Llorca et al. [19] reported enrichment on the functions related to the fatty acids metabolism and to sulfur and nitrogen metabolism in donor and maternal-milk groups in preterm infants.

Béghin et al. [20] reported lower butyric acid in the breastfed and prebiotic-supplemented formula-fed groups. Additionally, D-lactate and L-lactate were significantly higher in the formula-fed group.

Wang et al. [21], using functional analysis, showed that the bacterial genes that encoded glycine reductase, or were associated with periplasmic acid stress response in *Enterobacteria* acid resistance mechanisms, and L-fucose utilization functions were upregulated in breastfed infants, and at the same time a decreased expression of genes related to the methionine and valine degradation functions was observed compared to the formula group. This analysis used different gene ontology databases (InterPro2GO, SAMSA2) to find differential gene expression. The metatranscriptome data were analyzed and the reads that are involved in biological processes (e.g., catalytic activity) were obtained. There were genes differentially expressed depending on the feed type. Finally, Béghin et al. [20] showed that stool SIgA concentrations in the breastfed group were very high compared to the formula groups regardless of formula supplementation.

## 4. Conclusions

Overall, there is evidence to support that feed type modulates an infant’s microbiome constitution. The impact of feeding on term and preterm microbiota could have potential benefits on intestinal functionality, immune system, and metabolism, and probably pursuing the host for life.

In terms of the supplementation of the infant formulas, the included intervention studies showed that although Bifidobacteria-supplemented formula initially modulates the infant intestinal microbiome, this effect is not longstanding as it was observed that the Bifidobacteria ingested from formula failed to continue to remain in the infant intestinal system over time and the persistent colonization beyond intervention failed in all feeding groups.

Furthermore, providing a formula enriched with prebiotic GOS/FOS oligosaccharides to formula-fed infants with an established intestinal microflora results in an increased proportion of Bifidobacteria in the stools. This alteration of the intestinal bacterial communities has an impact on stool characteristics, such as a reduction in stool pH and a SCFA pattern containing a higher proportion of acetate and a lower proportion of propionate. The alterations induced by the addition of prebiotic oligosaccharides to infant formula lead the intestinal flora and its metabolic activity closer to that of breastfed infants.

This study has some limitations: (a) the included studies are not of optimal quality, due to their design (cohort studies) or other issues, such as no blinding or randomization reporting, though there are not many studies available; and (b) we searched a single database (PubMed), though a subsequent search in additional databases (Scopus and Google Scholar) did not produce more results.

In conclusion, these data provide a better understanding of the shaping of the gut microbiome in term and preterm infants depending on the feed type, which can be a valuable tool in the effort to improve infant formulas. They also open channels to explore the links between the gut microbiome–metabolome for biomarker detection and to identify important areas for future research.

## Figures and Tables

**Figure 1 children-09-00154-f001:**
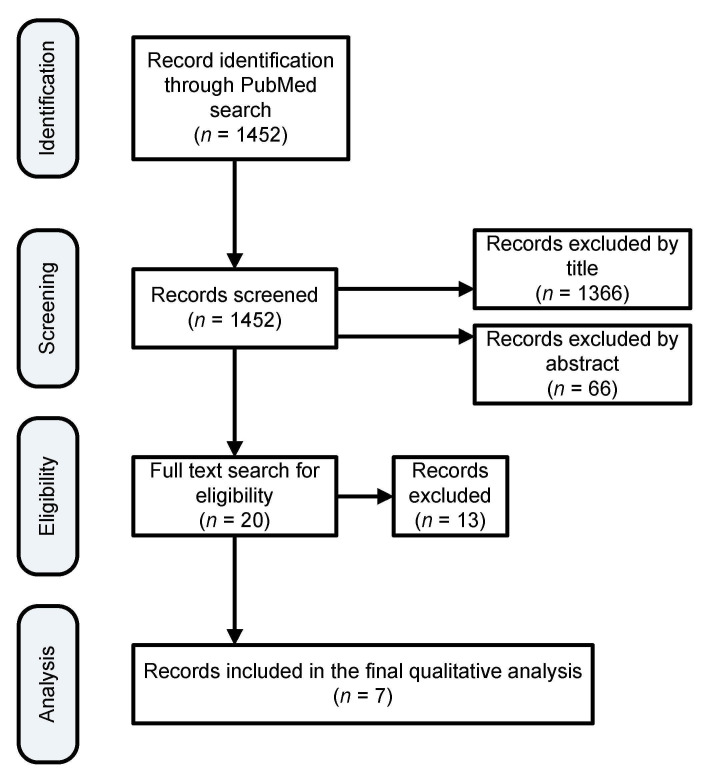
Search results filtration process to finalize the list of papers for qualitative analysis.

**Table 1 children-09-00154-t001:** PICO query components and final question.

PICO Component	Query Part
P(atient)	((infant*(Title/Abstract) OR infant*(MeSH Terms)) OR (neonat*(Title/Abstract) OR neonat*(MeSH Terms)))
I(ntervention)	((formul*(Title/Abstract) OR formul*(MeSH Terms)) OR (symbiot*(Title/Abstract) OR symbiot*(MeSH Terms)) OR (probiot*(Title/Abstract) OR probiot*(MeSH Terms)))
C(omparator, or control)	((breast milk(Title/Abstract) OR breast milk(MeSH Terms)) or (feed*(Title/Abstract) OR feed*(MeSH Terms)))
O(utcome)	((microb*(Title/Abstract) OR microb*(MeSH Terms)) OR (metabol*(Title/Abstract) OR metabol*(MeSH Terms)) or (fecal(Title/Abstract) OR fecal(MeSH Terms)) or (gut(Title/Abstract) OR gut(MeSH Terms)))
Τ(ime)	(“2000/01/01”(Publication Date): “3000”(Publication Date))
PICO Question	((infant*(Title/Abstract) OR infant*(MeSH Terms)) OR (neonat*(Title/Abstract) OR neonat*(MeSH Terms)))) AND ((formul*(Title/Abstract) OR formul*(MeSH Terms)) OR (symbiot*(Title/Abstract) OR symbiot*(MeSH Terms)) OR (probiot*(Title/Abstract) OR probiot*(MeSH Terms))) AND ((breast milk(Title/Abstract) OR breast milk(MeSH Terms)) or (feed*(Title/Abstract) OR feed*(MeSH Terms))) AND((microb*(Title/Abstract) OR microb*(MeSH Terms)) OR (metabol*(Title/Abstract) OR metabol*(MeSH Terms)) or (fecal(Title/Abstract) OR fecal(MeSH Terms)) or (gut(Title/Abstract) OR gut(MeSH Terms))) AND (“2000/01/01”(Publication Date): “3000”(Publication Date))

* is a wildcard character for example microb* stands for: microbes, microbiome, microbe etc.

**Table 2 children-09-00154-t002:** Characteristics of the included studies. BM: Breast Milk, DM/DHM: Donor human milk, BMF: Breast milk fortifier, FERM: Bioactive compounds, GOS: Galacto-oligosaccharides, FOS: Fructo-oligosaccharides, FGOS: Fructo & Galacto -oligosaccharides, IVH: Intraventricular hemorrhage, FISH: Fluorescence in situ hybridization.

	Study Design (Type, Consent)	Patient Characteristics (GA)	Observation Period	Feeding Types	Participants Size	Exclusion Criteria	Fecal Sample Processing	Primary Outcome
Cong et al., 2017 [17]	Secondary analysis of data from a prospective exploratory study, parental consent	Preterm 28–32 ^6/7^ weeks	30 days	BM, BM + DM, BM + Formula, DM, formula, DM + formula	38	Congenital abnormaitie, severe IVH, surgery, hx of prenatal drugs	16S rRNA gene amplicon sequencing	Gut microbial patterns associated with feeding type
Bazanella et al., 2017 [18]	Double-blind, randomized, placebo controlled, both parents consent	Term	12-month, (24-month)	BM, formula, Mixed	106	Preterm < 36 weeks, high-risk pregnancy, maternal chronic illness, antibiotics in the last 2months pregnancy	16S rRNA gene amplicon sequencing, metabolomics via UHPLC-MS	Fecal microbiota in the first year during Bifidobacteria supplementation, secondary:fecal metabolite profiling
Parra-Llorca et al., 2018 [19]	Prospective, observational, unicentric cohort study, parental consent	Preterm ≤ 32 weeks	12-month period	BM, DM, formula	69	Mixed brestfeeding not included, major malformations or surgery	16S rRNA gene amplicon sequencing	Impact of DHM on preterm microbiota
Béghin et al., 2020 [20]	Prospective, randomized, double-blind, controlled, multicentered (computer-generated randomization, parental consent	Term > 37 weeks	6 months	BM, formula, FERM, FERM/GOS/FOS, FGOS/FOS	350	Illness not included, congenital malformation, antibiotics, allergy	(FISH) Bacterial composition, Metabolic activity parameters (pH, SCFA, lactate)	SIgA concentration
Wang et al., 2020 [21]	Randomized controlled, parental consent	Late-preterm 32 ^0/7^–36 ^6/7^ weeks	17 days postnatally (2 samples 24 h apart) DNA and RNA extracted from fecal samples	BM, Formula	20	Exclusive BM (with BMF) or formula, All mothers received abx, No infant with positive blood culture (twins are acceptable)	16S rRNA gene amplicon sequencing, comparative metatranscriptomics (gene expression)	Alpha and beta diversity of gut bacterial composition (compare the composition and function of gut microbiome as related to the nutrional source in moderate-late preterm)
Li et al., 2020 [22]	Randomized controlled, parental consent	Term, 38 weeks	16 to 295 days	BM, Formula, mixed	77	No infants that received antibiotics and probiotics. History of chorioamnionitis and gestational diabetes	16S rRNA gene amplicon sequencing, Metabolomics by liquid chromatography-mass spectrometry	Gut microbiome composition and metabolites
Knol et al., 2005 [12]	randomized, double-blind, placebo controlled	Infants Age~7.7 weeks	7–8 weeks	Formula with prebiotic, standard formula, BM ©	68	Congenital abnormalities, allergy, antibiotics less than 2 weeks, formula with pre or probiotics less than a months before	FISH, gas chromatography(SCFA) pH electrode (pH), L-lactic detection kit (lactate)	Bifidobacteria, pH, SCFA, lactate

## Data Availability

Data are available from the corresponding author upon a reasonable request.

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
