# Peer review of "The Metagenomic and Metabolomic Profile of the Infantile Gut: Can They Be “Predicted” by the Feed Type?"

_children, 2022, doi:10.3390/children9020154_

Round 1

Reviewer 1 Report

Abstract:

  • “Trials compared the differences on metagenomics and metabolomics of infantile stool, aiming at investigating the beneficial effects of fortification of formula with synbiotics.” It is unclear which trials the authors refer to with this sentence (see comments below regarding the methods section).
  • Results of the literature screening should be summarized in the abstract (# of screened studies, # of studies eligible for inclusion)
  • “The evidence is sufficient towards the consensus”. The meaning of this sentence is not clear. I feel that we are still far from any “evidence” or “consensus” regarding optimal feeding strategies for young infants and regarding the exact mechanisms and directions through which feeding modulates gut microbiome. Please smooth this sentence, because this strong conclusion is not really supported by data from only seven, heterogeneous studies

Introduction

  • Line 57 and line 90, please check the references
  • The last sentence of the introduction would be more appropriate in the Methods section

Methods

  • Line 100-104, table 1: why the terms related to microbiome and metabolome were inserted in the patient section of the search string? Why did the authors not search other databases?
  • Line 111: please cite the specific guideline which was followed for this review and refer to the latest update of the PRISMA guideline (2020, not 2009 as in line 145)
  • Line 119-121: this is the same sentence which was unclear in the abstract. Which kind of trials were included? Only those assessing synbiotics vs. other types of diet?
  • In general, inclusion criteria for study selection should be reported more clearly, by identifying eligible study design (only RCTs?), characteristics of study populations (newborns? Infants? Both? Preterm and term?), intervention (formula + supplement vs. human milk?), outcome (microbiota? Metabolome? Both?)
  • Figure 1 (also lines 165-168): reasons for study exclusion should be detailed.
  • Line 156 and following: using GRADE approach would be useful for evaluating included studies. Please add this assessment to the text

Results

  • In the methods section, it is stated that the review would have aimed to evaluate RCTs, but in the first sentence of the results section different study designs are cited. Please clarify.
  • Line 174: which intervention? As stated above, it is unclear if this review aims at evaluating the effects of supplemented formula vs. other feed types or in general the effect of different diets (one vs. all the others) on infant gut microbiome and metabolome.
  • Lines 174-188: the summary of included studies is confounding. Given the low number of included studies, a synthetic description of each study in the main text should be more useful for the reader, rather than combining info from different studies about population, feeding, outcome and follow up. Furthermore, the exact composition of supplemented diets (probiotics, prebiotics, synbiotics) should be described in detail as it is certainly relevant for the study purpose. In this perspective, I think that generally referring to a comparison between “formula-fed” and breastfed infants is confounding, as different infant formula probably shape the gut microbiota differently and in "agreement" with their specific content.
  • Lines 216-217: the meaning of this sentence is unclear, as the results of the single studies have not been yet presented
  • The Results section about gut microbiota needs some improvement, as presented results should be put in the specific context of feeding type and study population. I think that results cannot be generalized, as it is well known that each kind of microbiota-modifying product (specific probiotic strain, prebiotics, and/or synbiotics) would shape the gut microbiota differently. Also the specific contribution of infants’ characteristics should be detailed (infants born via vaginal delivery or via C-section? Preterm or term infants? Late preterm or very preterm?). In line with previous comments, each study should be discussed separately, before being able to combine results of two or more studies, as the role of the cited confounding factors is likely to be relevant.
  • Following my previous comment, I would suggest the Authors to acknowledge that the results of only 7 studies, which are also heterogenous, should be analyzed cautiously and should not be generalized.

Conclusions

  • First sentence: see comment in the abstract
  • Is it there a missing Discussion section? I suggest rewriting the Results section by analyzing each study separately (see comments above) and provide a Discussion section in which results are discussed and heterogeneity/limitations are acknowledged

Author Response

 Abstract:

“The included trials compared the differences on metagenomics and metabolomics of       infantile stool, aiming at investigating the beneficial effects of fortification of formula with synbiotics.” It is unclear which trials the authors refer to with this sentence (see comments below regarding the methods section).

Authors’ actions: Changed. Please see track changes on the revised manuscript.

Results of the literature screening should be summarized in the abstract (# of screened studies, # of studies eligible for inclusion)

Authors’ actions:  Added a clarification:

Out of 1452 papers identified by the initial search, at the end 7 were selected for inclusion, following screening for eligibility. Eligibility was determined by closer examination for relevance of the title, abstract and subsequently full text

“The evidence is sufficient towards the consensus”. The meaning of this sentence is not clear. I feel that we are still far from any “evidence” or “consensus” regarding optimal feeding strategies for young infants and regarding the exact mechanisms and directions through which feeding modulates gut microbiome. Please smooth this sentence, because this strong conclusion is not really supported by data from only seven, heterogeneous studies

Authors’ actions: We changed the sentence . Please see track changes on the revised manuscript.

Introduction

Line 57 and line 90, please check the references

Authors’ actions: We have revised all references and corrected both missing information and formatting

The last sentence of the introduction would be more appropriate in the Methods section

Authors’ actions: Changed. Please see track changes on the revised manuscript.

Methods

Line 100-104, table 1: why the terms related to microbiome and metabolome were inserted in the patient section of the search string? Why did the authors not search other databases?

Authors’ actions: This was mistakenly referred in this section, it was already mentioned in the outcome section, we have revised the PICO query and performed corrections according to your comment, moreover we mention in the study limitations that we searched only one database and that from additional search in other databases, no more results were obtained.

Line 111: please cite the specific guideline which was followed for this review and refer to the latest update of the PRISMA guideline (2020, not 2009 as in line 145)

Authors’ actions: In the revised version is stated the most recent version of PRISMA guideline, thank you for pointing.

Line 119-121: this is the same sentence which was unclear in the abstract. Which kind of trials were included? Only those assessing synbiotics vs. other types of diet?

Authors’ actions:  The studies that were included are heterogeneous in terms of the types of formula that are used, so it could be either fortified or not, but they all compare and assess how the feed type modulates the gut micobiome and metabolome.

In general, inclusion criteria for study selection should be reported more clearly, by identifying eligible study design (only RCTs?), characteristics of study populations (newborns? Infants? Both? Preterm and term?), intervention (formula + supplement vs. human milk?), outcome (microbiota? Metabolome? Both?)

Authors’ actions:  We did not exclude studies initially just by the study design itself. Also the study populations included term or preterm neonates and infants up to 12 months of age (depending on the follow-up period). Intervention was basically formula, standard or fortified with pre-and/or- probiotics. The main outcome was both the composition of gut microbiota and metabolites.

Figure 1 (also lines 165-168): reasons for study exclusion should be detailed.

Authors’ actions:  The studies that were excluded were not relevant mainly because they used different biological specimens or study population other than neonates

Line 156 and following: using GRADE approach would be useful for evaluating included studies. Please add this assessment to the text

Authors’ actions: We added a separate appendix to evaluate the studies and the body of evidence according to GRADE methodology.

Results

In the methods section, it is stated that the review would have aimed to evaluate RCTs, but in the first sentence of the results section different study designs are cited. Please clarify.

Authors’ actions:  The studies that were included and their design are presented in the appendix table.

Line 174: which intervention? As stated above, it is unclear if this review aims at evaluating the effects of supplemented formula vs. other feed types or in general the effect of different diets (one vs. all the others) on infant gut microbiome and metabolome.

Authors’ actions:  The aim of this review is basically to present available data of the effect of the feed type on the infantile intestinal microbiota (microbial composition and metabolites) by DNA-sequencing and metabolome analysis of neonatal stool. This review also tries to evaluate whether clear differences, depending on feed type, exist.

Lines 174-188: the summary of included studies is confounding. Given the low number of included studies, a synthetic description of each study in the main text should be more useful for the reader, rather than combining info from different studies about population, feeding, outcome and follow up. Furthermore, the exact composition of supplemented diets (probiotics, prebiotics, synbiotics) should be described in detail as it is certainly relevant for the study purpose. In this perspective, I think that generally referring to a comparison between “formula-fed” and breastfed infants is confounding, as different infant formula probably shape the gut microbiota differently and in "agreement" with their specific content.

Authors’ actions:  Added to the text:

In one study, two groups of infants fed infant formula with  GOS/FOS, or control formula were evaluated in a randomized, double blind, placebo controlled intervention study. A breast-fed group was studied in parallel. At study onset and after 4 and 6 weeks, fecal samples were examined for the number of bifidobacteria, pH, short chain fatty acids and lactate.

The second study included preterm infants were recruited at birth and followed up for the first 30 days of life. Infants were classified into six groups (maternal milk [MOM],, human donor milk [HDM], Formula, MOM + HDM, MOM +Formula, and HDM + Formula). During postnatal days 0–10, 11–20, and 21–30, stool samples were collected daily.

The next study included infants that were observed from birth to 12 months postnatally and they were separated into three groups, breastfed, standard formula-fed and fortified-formula-fed (probiotics).

In the next study that was included, the effects of an infant formula containing both bioactive compounds (produced via a fermentation process) and prebiotics were investigated in healthy term infants. Three experimental formulas either containing a combination of these bioactive compounds (FERM) and prebiotic oligosaccharides (FERM/scGOS/lcFOS), prebiotic oligosaccharides (scGOS/lcFOS), or the bioactive compounds (FERM) were compared to a standard cow's milk-based control formula. As a reference, healthy term infants who were exclusively breastfed during the first six months of life were assessed.

In the next  study, the effects of feeding with mothers’ own breast milk (MBM) and formula on the initial composition and gene expression of gut bacteria in moderate–late preterm infants were investigated.

The next referring study, evaluated the impact of infant feed type from birth until 12 months of life depending on the duration of breastfeeding and the timing of formula introduction. Infants were classified into three groups, the exclusively breast-fed group, infants who had been only fed with formula and infants who had received both breast milk and formula prior to their fecal collection.

The last study had the objective to determine the impact of donor human milk upon preterm gut microbiota. The population were neonates <32 weeks of gestation and with a birth weight ≤1,500 g. Neonates were classified in three groups according to feeding practices consisting in their maternal milk, donor human milk or formula.

All the studies included, analyzed gut microbiota composition and metabolites of neonatal stool samples and compared the differences between different groups depending on the feed type.

Lines 216-217: the meaning of this sentence is unclear, as the results of the single studies have not been yet presented

Authors’ actions:  Deleted

The Results section about gut microbiota needs some improvement, as presented results should be put in the specific context of feeding type and study population. I think that results cannot be generalized, as it is well known that each kind of microbiota-modifying product (specific probiotic strain, prebiotics, and/or synbiotics) would shape the gut microbiota differently. Also the specific contribution of infants’ characteristics should be detailed (infants born via vaginal delivery or via C-section? Preterm or term infants? Late preterm or very preterm?). In line with previous comments, each study should be discussed separately, before being able to combine results of two or more studies, as the role of the cited confounding factors is likely to be relevant.

Authors’ actions:  

The population and feed type used in each study were described above. In this section there was an effort not exactly to generalize , but to describe and at the same time summarize the results of the studies by quoting the clearly common or conflicting observations and the identified similar conclusions.

Following my previous comment, I would suggest the Authors to acknowledge that the results of only 7 studies, which are also heterogenous, should be analyzed cautiously and should not be generalized.

Authors’ actions:  We do acknowledge that and tried to cautiously present and compare the results of these studies.

Conclusions

First sentence: see comment in the abstract

Authors’ actions:  Overall, there is evidence to support that feed type modulates the infants’ microbiome constitution.

Is it there a missing Discussion section? I suggest rewriting the Results section by analyzing each study separately (see comments above) and provide a Discussion section in which results are discussed and heterogeneity/limitations are acknowledged

Authors’ actions:  We think we comment at the limitations in the results section Line 212-221 and the appendix including the GRADE evaluation now also describes the heterogeneity/limitations.

Reviewer 2 Report

This is a very interesting and timely review of the literature related to modulation of the preterm and term infant gut microbiota based on prebiotic and probiotic supplementation. I find it quite interesting that, despite the intensive interest and research in this field, that too few studies were available to make a more quantitative meta-analysis, and I think the speaks to the importance of this review in guiding future research priorities. I have only minor comments.

Line-by-line comments:

Line 261. The Knoll et al. reference does not have a publication year.

 Line 293. The authors state that Proline increased the relative abundance of bifidobacteria. This implies causation. Is it not rather an association between bifidobacteria and proline levels?

Line 305. Please define "functional analysis". Is this metagenomics, or functional prediction with a tool like PieCrust?

Line 310. Is it presumed that the SIgA is coming from the breastmilk, or from the baby, or a combination of the two?

Author Response

This is a very interesting and timely review of the literature related to modulation of the preterm and term infant gut microbiota based on prebiotic and probiotic supplementation. I find it quite interesting that, despite the intensive interest and research in this field, that too few studies were available to make a more quantitative meta-analysis, and I think the speaks to the importance of this review in guiding future research priorities. I have only minor comments.

Line-by-line comments:

Line 261. The Knoll et al. reference does not have a publication year.

Authors’ actions: This is fixed in the revised manuscript

Line 293. The authors state that Proline increased the relative abundance of bifidobacteria. This implies causation. Is it not rather an association between bifidobacteria and proline levels?

Authors’ actions: Changed on the revised manuscript.

Line 305. Please define "functional analysis". Is this metagenomics, or functional prediction with a tool like PieCrust?

Authors’ actions: This analysis used different gene ontology databases (InterPro2GO , SAMSA2) to find differential gene expression. The metatranscriptome data was analyzed and the reads that are involved in biological processes (e.g. catalytic activity) were obtained. There were genes differentially expressed depending on the feed type.

Line 310. Is it presumed that the SIgA is coming from the breastmilk, or from the baby, or a combination of the two?

Authors’ actions: Breastfed babies secrete SIgA by gut mucosal tissue and this is important to the normal function of the GI tract as an immune barrier.

Round 2

Reviewer 1 Report

/